# Riociguat in Patients with CTEPH and Advanced Age and/or Comorbidities

**DOI:** 10.3390/jcm11041084

**Published:** 2022-02-18

**Authors:** Michaela Barnikel, Nikolaus Kneidinger, Paola Arnold, Andrea Waelde, Jürgen Behr, Katrin Milger

**Affiliations:** 1Department of Internal Medicine V, Ludwig Maximilian University of Munich, 81377 Munich, Germany; nikolaus.kneidinger@med.uni-muenchen.de (N.K.); paola.arnold@med.uni-muenchen.de (P.A.); andrea.waelde@med.uni-muenchen.de (A.W.); juergen.behr@med.uni-muenchen.de (J.B.); katrin.milger@med.uni-muenchen.de (K.M.); 2Comprehensive Pneumology Center (CPC-M), Member of the German Center for Lung Research (DZL), 81377 Munich, Germany

**Keywords:** hemodynamics, efficacy, tolerability

## Abstract

Riociguat is licensed for the therapy of inoperable chronic thromboembolic pulmonary hypertension (CTEPH). We aimed to investigate whether age and comorbidities influence its tolerability and efficacy. Retrospectively, we analyzed data of tolerability, non-invasive, and invasive efficacy at baseline and follow up (FU) of all patients with CTEPH treated with riociguat at the Department of Internal Medicine V, University of Munich (*n* = 47), grouping patients according to age (<65 versus 65–79 versus ≥80 years) and risk factors for heart failure with preserved ejection fraction (HFpEF) (<2 versus ≥2 risk factors). During dose titration patients >80 years reported side effects more frequently (40%) than the other age groups (23% and 21% for patients <65 years and patients 65–79, respectively). Cessation of riociguat was rare and occurred independent of age. When looking at the total cohort of 47 patients, three patients stopped therapy and three patients had a reduced maintenance dosage, while 41/47 (87%) and all octogenarians reached the highest maintenance dosage of 7.5 mg/d. The frequency of any side effect was similar in patients in both risk factor groups, and hypotension was only observed in those with <2 risk factors. Parameters of efficacy improved significantly under riociguat treatment. Improvement in 6-min walk distance (6 mwd), N-terminal pro brain natriuretic peptide (Nt-proBNP) and hemodynamics did not differ between age or risk factor groups. In this small real-life cohort, riociguat was well-tolerated and effective in advanced age and risk factors for HFpEF.

## 1. Introduction

Pulmonary hypertension (PH) is a progressive, life-threatening disease, which is classified into five groups based on the cause, pathologic findings, and hemodynamic characteristics [1]. Treatments for pulmonary arterial hypertension (PAH, group 1) target three pathways: the prostacyclin pathway, the endothelin pathway, and the nitric oxide (NO)—soluble guanylate cyclase (sGC)—cyclic guanosine monophosphate (cGMP) pathway [2].

Riociguat is an oral stimulator of the soluble guanyl-cyclase (sGC stimulator) [3] approved for the treatment of adults with PAH and adults with chronic thromboembolic hypertension (CTEPH) who are inoperable or have persistent/recurrent PH after pulmonary endarterectomies (PEA) [4,5,6] but not for patients with PH due to left heart or lung disease [7,8,9]. PEA is the gold standard treatment for CTEPH, as it is the only potentially curative treatment [10]. However, up to 40% of CTEPH patients are ineligible for PEA for various reasons, such as distal lesions and severe comorbidities [11]. For these patients, riociguat is currently the only approved oral therapy and used as the first line medical treatment, although other agents are being studied [12].

In patients with PAH who have cardiopulmonary comorbidities and/or advanced age, monotherapy with phosphodiesterase-5 (PDE-5)—inhibitor is often chosen. Other therapies, including riociguat, are used less frequently due to the concern of increased side effects in this population. The three-times-a-day dosing might be another obstacle to choosing riociguat. Of note, patients over 80 years were generally excluded from the licencing trials with riociguat [4,5] and recent trials in PAH have also excluded patients with ≥2 risk factors for heart failure with preserved ejection fraction (HFpEF) [13]. Frequent side effects of riociguat include hypotension and dizziness as well as gastroesophageal reflux and these might be of particular concern in older patients with comorbidities.

We aimed to investigate whether age and comorbidities influence the tolerability and efficacy of riociguat. Thus, we analyzed all CTEPH-patients treated at our center with riociguat according to comorbidities and age.

## 2. Materials and Methods

### 2.1. Study Design

This was a single-center, retrospective study at the Department of Internal Medicine V, University of Munich. Data were collected corresponding to the Good Clinical Practice. The study was approved by the local ethics committee (No 19-883).

### 2.2. Patients

We analyzed all patients with the diagnosis of CTEPH, verified by right heart catheterization (RHC) as mPAP ≥ 25 mmHg and PAWP ≤ 15 mmHg and corresponding imaging, who were initiated on riociguat at our department since its approval in 2014 up to 2019. At diagnosis all patients were assessed by RHC, ventilation perfusion scan, computed tomography angiography (CTA) pulmonalis and angiography of the pulmonary arteries. Results were reviewed at an interdisciplinary board consisting of an interventional radiologist, specialized heart surgeon, pulmonologist, and cardiologist. Technical inoperability was defined if the CTEPH board concluded that lesions were too peripheral to be addressed by surgery. Medical inoperability was defined if the board advised against operation due to comorbidities or severely impaired cardiac output presenting a high anesthesiologic risk for the operation. Optimization of fluid status using diuretics prior to RHC and start of riociguat (baseline) was routinely performed. Patients who had residual PH (mPAP > 25) after pulmonary endarterectomy and/or balloon angioplasty and received riociguat for treatment of residual PH were also included. All patients were adults and received either no PH-therapy before riociguat or had been on a stable treatment with endothelin-receptor antagonist (ERA) or phosphodiesterase-5 inhibitors (PDE5-inhibitors) for at least 3 months prior to initiation of riociguat. In case of prior treatment with PDE5-inhibitors, a switch to riociguat was performed as a combination of PDE5-inhibitors and riociguat is contraindicated. Patients were grouped according to their age at initiation of treatment with riociguat, divided into cohorts < 65 years, 65–79 years and ≥80 years. Furthermore, patients were classified based on comorbidities and risk factors for heart failure with preserved ejection fraction (HFpEF) (<2 risk factors versus ≥2 risk factors). These risk factors included diabetes mellitus (DM), atrial fibrillation (AF), coronary heart disease (CHD), arterial hypertension (AH), and obesity with a body mass index (BMI) ≥30 kg/m^2^.

### 2.3. Procedures

Therapy with riociguat was initiated with 0.5 mg three times daily or 1.0 mg three times daily as judged by the treating physician. Dosage was uptitrated individually to a maximum of 2.5 mg three times daily. Uptitration was administered biweekly as long as no unmanageable side effects occurred. The maintenance dosage was achieved when no further increase was possible or the maximum of 2.5 mg three times daily was reached. In case of switching from a PDE5-inhibitor to riociguat, a drug-free interval of 24 h for sildenafil and 48 h for tadalafil was met. Dosage titration was supported by an external patient service who interviewed the patients’ health status and documented the occurrence of side effects in a ten-day interval. Further, blood pressure was measured at home every morning by patients themselves and transmitted electronically to the treating physician. Hypotension was defined as systolic blood pressure of ≤90 mmHg. If so, reducing dosage took place only in case of accompanying clinical symptoms of hypotension such as lightheadedness. Mild side effects were treated with supportive therapy using antiemetic, antidiarrheal and/or analgetic drugs. In case of severe side effects, the patients were requested to consult the physician to evaluate how to deal with the further dosage of riociguat.

Efficacy data were collected at baseline and follow up (FU). Baseline was defined as time of stable or no PH-therapy before starting riociguat. FU was defined as time of first RHC after initiating riociguat.

Data of efficacy were collected according to the risk assessment for PH and included the following: Non-invasive parameters including the World Health Organization functional class (WHO-FC), brain natriuretic peptide (Nt-proBNP), 6-min walk distance (6 mwd), partial pressure of oxygen of arterialized capillary blood (p_a_O_2_), tricuspid annular plane systolic excursion (TAPSE), right atrial area (RAA), as well as invasive parameters involving mean pulmonary arterial pressure (mPAP), mean right atrial pressure (mRAP), cardiac index (CI), pulmonary vascular resistance (PVR), mixed venous saturation (SvO_2_%) and the pulmonary capillary wedge pressure (PAWP). Functional class (FC) score was calculated as the sum of all FC class numbers in each group and change of FC score was indicated relative to group size in order to allow comparisons between groups. In detail the formula is: FC score = FC class (patient 1) + FC class (patient 2) + …... *n* (patients). Delta FC score = (FC score after riociguat)−(FC score before riociguat). Delta FC score % of *n* = (Delta FC score/*n*) ×100.

All medical examinations were performed as required by applicable guidelines; cardiac output was measured by thermodilution.

### 2.4. Statistical Analysis

Parameters are given as mean ± standard deviation (SD) if normally distributed, otherwise as median with range. To identify differences in continuous variables, paired *t*-test (parametric) or Wilcoxon-test (non-parametric) were used for paired variables. Comparisons of continuous but unpaired variables were performed using Mann-Whitney-U test (2 groups) and ANOVA (>2 groups), as appropriate. Chi square test and Fisher’s exact test compared categorical variables. Values of *p* < 0.05 were considered significant. All statistical analyses were performed with Graph Pad Prism version 8.3.0. With analysis of multiple parameters, statistical analysis is exploratory as no correction for multiple testing was applied.

## 3. Results

### 3.1. Study Cohort

During the studied timeframe, 55 patients with CTEPH received riociguat, 8 patients were excluded due to missing data, and 47 patients were included in the tolerability analysis (Figure 1). Three patients discontinued therapy with riociguat because of intolerable side effects already at low dosage before FU. These three cases are discussed below. Apart from these three patients, FU-data for analysis of efficacy are available for 44 patients. Hereof 12 patients (27%) were <65 years, 23 patients (52%) aged 65–79 years, and 9 patients (21%) ≥ 80 years. At least one of the defined cardiopulmonary comorbidities was found in 40 subjects (91%), while 19 subjects (40%) had two or more risk factors for HFpEF. Patients <65 years had comorbid arterial hypertension less frequently than older patients and baseline six mwd and TAPSE decreased with increasing age. There were no differences in baseline hemodynamic parameters (Table 1).

### 3.2. Safety

In our total cohort, 12 patients (26%) reported at least one side effect under titration with riociguat. When looking at the total cohort of 47 patients, three patients stopped therapy and three patients had a reduced maintenance dosage, 41/47 (87%) reached the highest maintenance dosage of 7.5 mg/d. During dose titration patients < 65 years and patients 65–79 years reported side effects less frequently (23% and 21%, respectively) than those ≥ 80 years (40%). A reduced maintenance dosage of less than 3 × 2.5 mg/d as a consequence of side effects was applied in one (8%), two (8%) and zero (0%) patients respectively (Figure 2A). Furthermore, the occurrence of any side effect did not differ significantly between patients with ≥2 risk factors for HFpEF and patients < 2 risk factors (21% and 29%, respectively, *p* = 0.72). Likewise, the frequency of reduced dosage was similar (5% and 7%, respectively, *p* > 0.99) (Figure 2B).

Figure 3 illustrates the characteristics of side effects under riociguat. Noticeable, hypotension was observed only in patients with fewer than two risk factors for HFpEF and was numerically more frequent in patients without arterial hypertension (16%, 3/19) than those with comorbid arterial hypertension (4%, 1/25). No other association of side effects with specific comorbidities were found. All observed side effects were within the spectrum of the known safety profile of riociguat. There was no case of hemoptysis.

As mentioned above, three patients discontinued therapy with riociguat because of unmanageable side effects already at a low dosage (Figure 1). First, this was a 49-year-old man under endothelin-receptor-antagonist, who received riociguat as add-on therapy. He reported thoracic pain and thoracic paresthesia and quit riociguat without physician’s consultation. After quitting riociguat, a therapy with phosphodiesterase-5-inhibitor was started and well-tolerated. Second, a 73-year-old woman suffered from symptomatic hypotension at a dosage of 0.5 mg riociguat three times daily. Afterwards, she received an endothelin-receptor-antagonist, and later additional therapy with phosphodiesterase-5-inhibitor with an acceptable tolerability. Third, an 84-year-old woman reported various side effects including nausea, dizziness, palpitation, fatigue and cough at a maximum dosage of 1.0 mg three times daily. She was successfully switched to an endothelin-receptor-antagonist.

### 3.3. Efficacy

FU—Data were collected after a mean of 337 days. Considering the total cohort, there was a significant improvement of hemodynamic and clinical parameters under therapy with riociguat (Table 2). In detail this improvement in CI (baseline 2.6 ± 0.6 L/min/m^2^; FU 3.0 ± 0.7 L/min/m^2^; Δ0.4 ± 0.8 L/min/m^2^; *p* = 0.0006), mPAP (baseline 45 ± 12 mmHg; FU 39 ± 9 mmHg; Δ−6 ± 9 mmHg; *p* = 0.003), PVR (baseline 8 ± 4 WE; FU 5 ± 2WE; Δ−3 ± 3WE, *p* < 0.0001), SvO_2_ (baseline 63 ± 6; FU 66 ± 6%, Δ3 ± 6%, *p* = 0.0112), Nt-proBNP (baseline 1260 (47; 14,429) pg/mL; FU 697 (58; 5115) pg/mL; Δ−336 (−9314;1668), *p* = 0.0039), 6 mwd (baseline 316 ± 121 m; FU 345 ± 114 m; Δ29 ± 63 m; *p* = 0.0152) and TAPSE (baseline 17 ± 4 mm; FU 20 ± 4 mm; Δ2 ± 4 mm; *p* = 0.0059), while p_a_O_2_ deteriorated (baseline 58 ± 8 mmHg; FU 55 ± 8 mmHg; Δ−3 ± 6 mmHg; *p* = 0.0017). Analysing the correlation between ∆mPAP (delta mean pulmonary arterial pressure = mPAP at initial right heart catheter—mPAP at follow-up catheter) and ∆mAP (delta mean arterial pressure = mAP at initial catheter—mAP at follow-up catheter), there was a trend towards a weak correlation in the total cohort (r = 0.29, *p* = 0.07).

The FC score (sum of the FC classes) of our patients improved by six points (out of 10 patients, 60%), two points (out of 22 patients, 10%) and five points (out of nine patients, 55%) in the age groups <65 years, 65–79 years, >80 years, respectively, while improvement of 6 mwd, Nt-proBNP, and hemodynamics did not differ between age groups (Table 3).

In patients with <2 risk factors for HFpEF, the FC score improved by 13 points (out of 24 patients, 54%), as opposed to two points (out of 17 patients; 12%) in patients with ≥2 risk factors for HFpEF. Mean improvement in 6 mwd did not differ significantly, albeit numerically greater improvement was found in those with ≥2 risk factors. Improvement in Nt-proBNP and hemodynamics was similar between risk factor groups (Table 4).

## 4. Discussion

We aimed to analyze the efficacy and safety of riociguat in patients with CTEPH according to their age and risk factors for HFpEF. Parameters of the risk assessment improved in all groups under therapy with riociguat. This improvement did not differ significantly between groups for most of the studied parameters, but we detected differences in certain parameters of tolerability and efficacy according to age or HFpEF risk factor group.

First, reporting of any side effect during dose titration occurred in numerically higher percentage in octogenarians than in the other age groups. However, this group was small, and it did not translate into a higher frequency of stopping treatment or reduced maintenance dosage. This underlines that also in this age group, side effects are mostly mild, manageable, or transient. The occurrence of any side effect did not differ between risk factor groups, but hypotension was only observed in patients with <2 risk factors. Probably, the high prevalence of arterial hypertension in patients with ≥2 risk factors is the reason why this group was not affected by hypotension. Other comorbidities like diabetes mellitus, atrial fibrillation, high body mass index, or coronary heart disease were not associated with increased or decreased frequency of side effects, or occurrence of specific side effects like diarrhea, dizziness, nausea, or headache in our cohort. Improvement in WHO-FC class was larger in patients <65 years than in the other age groups. Likewise, patients with <2 risk factors for HFpEF had greater improvement in WHO class than those with more risk factors. However, in both cases such differences were not observed in 6 mwd, Nt-proBNP, or hemodynamics. A possible explanation could be that the impairment of functional class is multifactorial in patients of advanced age and those with comorbidities and thus more difficult to improve with vasoactive therapy alone.

The PATENT-study found a significant increase of exercise capacity and hemodynamics under riociguat in patients with pulmonary arterial hypertension [4]. The CHEST–study evidenced similar results for patients with CTEPH who were deemed ineligible for surgery or who had persistent or recurrent pulmonary hypertension after undergoing pulmonary endarterectomy [5]. Both trials excluded patients older than 80 years, even though in clinical practice patients with CTEPH are often of older age, as illustrated by our study. Currently riociguat remains the only oral treatment for patients with CTEPH [14] whereas PAH PDE5-inhibitors are also licensed and often preferred as the first line treatment in patients with advanced age or comorbidities assuming better tolerability [15]. Our results showed minor differences in certain tolerability and efficacy parameters in patients with CTEPH according to age or comorbidity, but these differences do not justify withholding the treatment from these patients. Detailed considerations and ideas in the management of older patients with PAH are discussed in a review of Sitbon et al. [16]. It recommends the assessment of goals, expectations and treatment tolerability in older patients on an individual basis [16]. Our data support the view that riociguat should be considered as a treatment option in these patients; as side effects may occur more frequently in older patients, information about side effects and their management as well as monitoring is of importance.

Switching from PDE5-inhibitor to riociguat in PAH was shown to improve outcomes in the RESPITE and REPLACE trials [13,17]. However, patients over 75 years and those with risk factors for HFpEF have again been excluded from the trials. Our present data in patients with CTEPH indicate that these patient groups may experience similar benefit from riociguat as other groups and thus should also be considered for this therapy. Further trials need to analyze tolerability of riociguat in patients with PAH and advanced age or comorbidities, as comorbidities can mask symptoms of PAH or increase the difficulty of evaluating disease progression and treatment effects by confounding prognostic assessment [18]. Underlying cardiac comorbidities including risk factors for HFpEF may complicate the discrimination of PH etiology. However, our results showed no gross differences in the profile of safety or efficacy of riociguat between patients with ≥2 risk factors or patients with <2 risk factors of HFpEF.

While 26% of our cohort reported any side effect under titration with riociguat, 87% tolerated the maximum dose at the end of an individual uptitration. Thus, the percentage of patients reaching the maximum maintenance dosage compared was higher compared to the phase three studies of riociguat, which reported the maximum dosage of riociguat for 75% of patients with PAH and 77% of patients with CTEPH [4,5], respectively, and 83% in another long-term study cohort of CTEPH [19]. The documented side effects were similar in all studies including this one and correspond to the NO-sGC-cGMP pathway.

While in this study hemodynamic parameters improved, there was a small deterioration of p_a_O_2_ under riociguat (mean decrease to baseline in the total cohort of 3 mmHg, *p* = 0.0013). A decrease of p_a_O_2_ was also found in the phase three trial of riociguat in PAH [20], albeit to a lesser extent, but studies in CTEPH largely omitted reporting p_a_O_2_. Increased ventilation-perfusion mismatch under riociguat and thus worsening of blood oxygenation might be the mechanism behind this finding.

This study should be interpreted in view of its strengths and limitation. The strengths include a real-life cohort that included patients who are often excluded from clinical trials and a very thorough characterization of the patients and their comorbidities. The main limitation is its retrospective, single-center design with a limited number of patients. In order not to overlook small differences due to sample size, we have also reported qualitative differences and used exploratory analysis in this manuscript. Second, we only analyzed patients with the diagnosis of CTEPH but not with PAH, in whom the question of age and comorbidities is also meaningful as there is a variety of drugs in the medical treatment for PAH and therefore the option of choice.

## 5. Conclusions

In conclusion, in our patients with CTEPH, therapy with riociguat was effective and well-tolerated in patients of advanced age or risk factors for HFpEF.

## Figures and Tables

**Figure 1 jcm-11-01084-f001:**
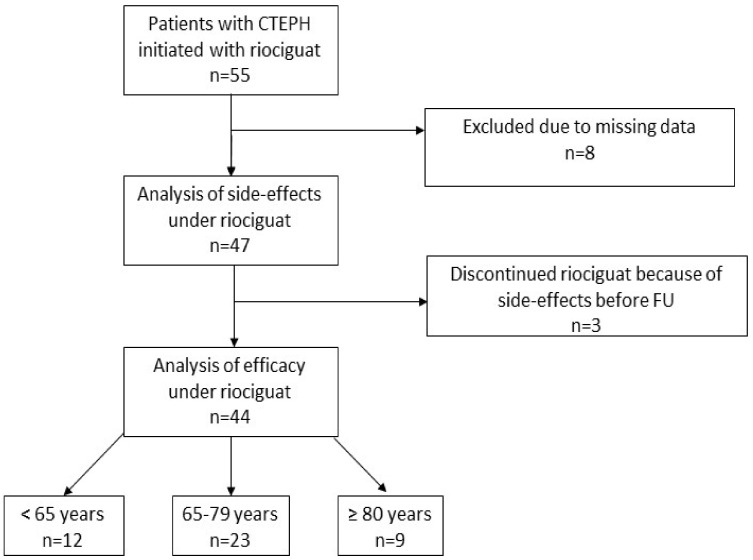
Study cohort. Abbreviation: Chronic thromboembolic pulmonary hypertension (CTEPH), follow-up (FU).

**Figure 2 jcm-11-01084-f002:**
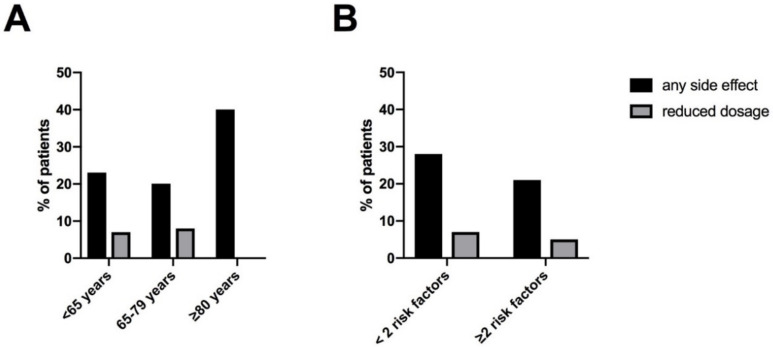
Side effects and maintenance dosage of riociguat: (**A**) according to age (**B**) according to risk factors for HFpEF (BMI ≥ 30 kg/m^2^, AH, DM, CHD, AF). Abbreviation: atrial fibrillation (AF), arterial hypertension (AH), body mass index (BMI), coronary heart disease (CHD), diabetes mellitus (DM), heart failure with preserved ejection fraction (HFpEF).

**Figure 3 jcm-11-01084-f003:**
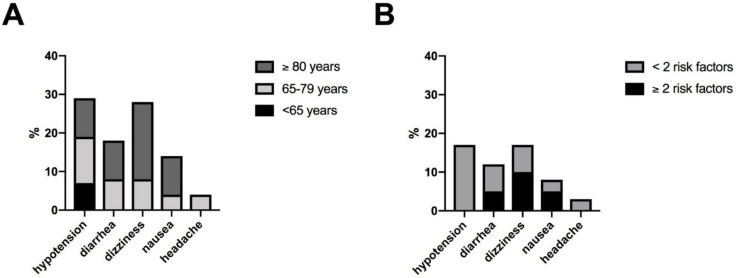
Characteristics of side effects: (**A**) according to age (**B**) according to risk factors for HFpEF (BMI ≥ 30 kg/m^2^, AH, DM, CHD, AF). Abbreviation: atrial fibrillation (AF), arterial hypertension (AH), body mass index (BMI), coronary heart disease (CHD), diabetes mellitus (DM), heart failure with preserved ejection fraction (HFpEF).

**Table 1 jcm-11-01084-t001:** Baseline characteristics.

	Total	<65 Years	65–79 Years	≥80 Years	*p*-Value
	*n* = 47	*n* = 13	*n* = 24	*n* = 10	
Female, *n* (%)	29 (62)	6 (46)	18 (75)	5 (50)	
Age (years)	69 ± 14	51 ± 12	73 ± 5	82 ± 2	<0.0001
BMI (kg/m^2^)	26 ± 5	25 ± 5	28 ± 6	24 ± 4	0.10
BMI ≥ 30 kg/m^2^, *n* (%)	12 (26)	3 (23)	7 (30)	0 (0)	0.16
Surgical/Interventional treatment prior baseline and riociguat initiation, *n* (%)					
Pulmonary endarterectmoty (PEA)	6 (13)	1 (8)	5 (21)	0 (0)	
No PEA	41 (87)	12 (92)	19 (79)	10 (100)	
Technically inoperable	17 (36)	9 (69)	6 (25)	2 (20)	
Medically inoperable	14 (30)	3 (23)	7 (29)	4 (40)	
Surgery refused by patient	8 (17)	0 (0)	6 (25)	2 (20)	
No surgery for unknown reason	2 (4)	0 (0)	0 (0)	2 (20)	
No PEA, but balloon angioplasty	1 (2)	0 (0)	1 (4)	0 (0)	
Prior PH medication, *n* (%)					
None	39 (83)	10 (77)	20 (83)	9 (90)	0.71
ERA add on riociguat	2 (4)	2 (15)	0 (0)	0 (0)	0.06
Switch PDE 5-inhibitor to riociguat	6 (13)	2 (15)	3 (13)	1 (1)	0.93
Switch ERA to riociguat	1 (2)	0 (0)	1 (4)	0 (0)	0.98
Comorbidities, *n* (%)					
COPD	6 (13)	1 (8)	3 (13)	2 (20)	0.77
ILD	0 (0)	0 (0)	0 (0)	0 (0)	-
Diabetes mellitus	4 (9)	0 (0)	3 (13)	1 (10)	0.42
OSAS	3 (6)	1 (8)	2 (8)	0 (0)	0.65
Atrial fibrillation	14 (30)	3 (23)	10 (42)	1 (10)	0.15
Coronary artery disease	9 (19)	3 (23)	4 (16)	2 (20)	0.89
Arterial hypertension	25 (53)	2 (15)	16 (67)	7 (70)	0.0057
WHO-FC, *n* (%)	*n* = 44	*n* = 12	*n* = 22		
II	9 (20)	3 (25)	5 (23)	1 (10)	
III	33 (75)	8 (67)	16 (73)	9 (90)	
IV	2 (5)	1 (8)	1 (4)	0 (0)	
Clinical parameters	*n* = 37–45	*n* = 9–13	*n* = 17–23		
Nt-proBNP (pg/mL)	1252 (47; 14,429)	1260 (148; 7043)	1067 (47; 5317)	2313 (158; 14,428)	0.45
6 mwd (m)	321 ± 117	404 ± 93	302 ± 113	271 ± 110	0.0211
p_a_O_2_ (mmHg)	58 ± 9	59 ± 12	58 ± 7	56 ± 8	0.69
Echocardiography	*n* = 33	*n* = 10	*n* = 15	*n* = 8	
TAPSE (mm)	17 ± 4	16 ± 6	19 ± 3	14 ± 4	0.0301
RHC	*n* = 39–47	*n* = 11–13	*n* = 19–24	*n* = 9–10	
mPAP (mmHg)	45 ± 12	49 ± 13	42 ± 11	48 ± 13	0.21
mRAP (mmHg)	8 ± 4	8 ± 4	7 ± 4	9 ± 4	0.43
CI (L/min/m^2^)	2.6 ± 0.7	2.5 ± 0.6	2.8 ± 0.7	2.6 ± 0.6	0.51
PVR (WE)	8 ± 4	9 ± 4	8 ± 5	9 ± 4	0.73
SvO_2_ (%)	63 ± 7	63 ± 7	64 ± 4	61 ± 7	0.49
PAWP (mmHg)	10 ± 3	9 ± 3	9 ± 4	10 ± 4	0.72

Abbreviations: Body mass index (BMI), cardiac index (CI), chronic obstructive pulmonary disease (COPD), endothelin receptor antagonist (ERA), interstitial lung disease (ILD), mean pulmonary arterial pressure (mPAP), mean right atrial pressure (mRAP), N-terminal pro brain natriuretic peptide (Nt-proBNP), obstructive sleep apnoe syndrome (OSAS), pulmonary arterial wedge pressure (PAWP), phosphodiesterase 5 inhibitor (PDE 5-inhibitor), pulmonary endarterectomy (PEA), pulmonary hypertension (PH), oxygen partial pressure from arterialized capillary blood (p_a_O_2_), pulmonary vascular resistance (PVR), right atrial area (RAA), central venous oxygen saturation (SvO_2_), tricuspid annular plane systolic excursion (TAPSE), World Health Organization—functional class (WHO-FC), 6 min walk distance (6 mwd).

**Table 2 jcm-11-01084-t002:** Follow up of the ungrouped cohort, *n* = 44.

	Baseline	Follow-Up	∆	*p*-Value
RHC, *n* = 40				
mPAP (mmHg)	45 ± 12	39 ± 9	−6 ± 9	0.003
mRAP (mmHg)	8 ± 4	7 ± 3	−1 ± 4	0.10
CI (L/min/m^2^)	2.6 ± 0.6	3.0 ± 0.7	0.4 ± 0.8	0.006
PVR (WE)	8 ± 4	5 ± 2	−3 ± 3	<0.0001
SvO_2_ (%)	63 ± 6	66 ± 6	3 ± 6	0.0112
PAWP (mmHg)	9 ± 4	10 ± 4	0.9 ± 4	0.20
WHO-FC, *n* = 41				
I	0 (0)	3 (8)	3	
II	9 (20)	17 (41)	8	
III	29 (71)	19 (46)	−10	
IV	3 (9)	2 (5)	−1	
FC Score	117	102	−15	
Clinical parameters				
Nt-proBNP (pg/mL), *n* = 37; (range)	1260 (47; 14,429)	697 (58; 5115)	−336 (−9314; 1668)	0.0039
6 mwd (m), *n* = 32	316 ± 121	345 ± 114	29 ± 63	0.0152
p_a_O_2_ (mmHg), *n* = 39	58 ± 8	55 ± 8	−3 ± 6	0.0013
Echocardiography				
TAPSE (mm), *n* = 25	17 ± 4	20 ± 4	2 ± 4	0.0059
RAA (cm^2^), *n* = 18	26 ± 6	24 ± 5	−2 ± 6	0.19

Abbreviations: body mass index (BMI), cardiac index (CI), mean pulmonary arterial pressure (mPAP), mean right atrial pressure (mRAP), n-terminal pro brain natriuretic peptide (Nt-proBNP), pulmonary arterial wedge pressure (PAWP), oxygen partial pressure of arterialized capillary blood (p_a_O_2_), pulmonary vascular resistance (PVR), right atrial area (RAA), right heart catheterization (RHC), central venous oxygen saturation (SvO_2_), tricuspid annular plane systolic excursion (TAPSE), World Health Organization—functional class (WHO-FC), 6 min walk distance (6 mwd).

**Table 3 jcm-11-01084-t003:** Follow up according to age cohort.

	<65 Years	65–79 Years	≥80 Years	*p*-Value
	*n* = 12	*n* = 23	*n* = 9	
RHC	*n* = 11–12	*n* = 17–23		
∆mPAP (mmHg)	−10 ± 14	−4 ± 8	−10 ± 11	0.19
∆mRAP (mmHg)	2 ± 2	−1 ± 4	−2 ± 4	0.42
∆CI (L/min/m^2^)	0.3 ± 0.6	0.3 ± 0.9	0.6 ± 0.7	0.56
∆PVR (WE)	−3 ± 3	−2 ± 3	−4 ± 3	0.45
∆SvO_2_ (%)	2 ± 6	2 ± 5	5 ± 6	0.39
∆PAWP (mmHg)	2 ± 6	1 ± 4	−1 ± 5	0.43
WHO-FC, *n* (%)	*n* = 10	*n* = 22	*n* = 9	
∆I	2	1	0	
∆II	2	1	5	
∆III	−4	−1	−5	
∆IV	0	−1	0	
∆ FC Score (% of *n*)	−6 (60%)	−2 (9%)	−5 (55%)	
Clinical parameters	*n* = 7–12	*n* = 6–22	*n* = 8–9	
∆Nt-proBNP (pg/mL)	−1247 (1063; 3074)	−43 (4588; 1668)	−1015 (−9314; 588)	0.07
∆6 mwd (m)	22 ± 71	31 ± 65	30 ± 60	0.96
∆p_a_O_2_ (mmHg)	−2 ± 7	−3 ± 14	−1 ± 4	0.89
Echocardiography	*n* = 7–9	*n* = 11–13	*n* = 5–6	
∆TAPSE (mm)	3 ± 4	2 ± 4	1 ± 3	0.63
∆RAA (cm^2^)	0 ± 0	−3 ± 7	−1 ± 4	0.71

Abbreviations: body mass index (BMI), cardiac index (CI), mean pulmonary arterial pressure (mPAP), mean right atrial pressure (mRAP), n-terminal pro brain natriuretic peptide (Nt-proBNP), pulmonary arterial wedge pressure (PAWP), oxygen partial pressure (p_a_O_2_), pulmonary vascular resistance (PVR), right atrial area (RAA), right heart catheterization (RHC), central venous oxygen saturation (SvO_2_), tricuspid annular plane systolic excursion (TAPSE), World Health Organization—functional class (WHO-FC), 6 min walk distance (6 mwd).

**Table 4 jcm-11-01084-t004:** Follow up according risk factors for HFpEF.

	<2 Risk Factors	≥2 Risk Factors	*p*-Value
	*n* = 26	*n* = 18	
RHC	*n* = 18–23	*n* = 13–17	
∆mPAP (mmHg)	−6 ± 10	−6 ± 9	0.98
∆mRAP (mmHg)	−1 ± 3	−2 ± 4	0.37
∆CI (L/min/m^2^)	0.4 ± 0.9	0.3 ± 0.6	0.97
∆PVR (WE)	−3 ± 3	−3 ± 2	0.83
∆SvO_2_ (%)	4 ± 6	1 ± 6	0.11
∆PAWP (mmHg)	1 ± 5	1 ± 5	0.71
WHO-FC, *n* (%)	*n* = 24	*n* = 17	
∆I	3	0	
∆II	5	3	
∆III	−6	−4	
∆IV	−2	1	
∆FC score (% of *n*)	−13 (54%)	−2 (12%)	
Clinical parameters	*n* = 21–25	*n* = 11–14	
∆Nt-proBNP (pg/mL)	−400 (−9314; 701)	−90 (−4588; 1668)	0.39
∆6 mwd (m)	19 ± 60	47 ± 68	0.23
∆p_a_O_2_ (mmHg)	−4 ± 6	−4 ± 6	0.73
Echocardiography	*n* = 16–25	*n* = 9	
∆TAPSE (mm)	3 ± 4	1 ± 3	0.09
∆RAA (cm^2^)	−2 ± 4	−2 ± 8	0.44

Abbreviations: body mass index (BMI), cardiac index (CI), heart failure with preserved ejection fraction (HFpEF), mean pulmonary arterial pressure (mPAP), mean right atrial pressure (mRAP), n-terminal pro brain natriuretic peptide (Nt-proBNP), pulmonary arterial wedge pressure (PAWP), oxygen partial pressure (p_a_O_2_), pulmonary vascular resistance (PVR), right atrial area (RAA), right heart catheterization (RHC), central venous oxygen saturation (SvO_2_), tricuspid annular plane systolic excursion (TAPSE), World Health Organization—functional class (WHO-FC), 6 min walk distance (6 mwd).

## Data Availability

The data presented in this study are available on request from the corresponding author. The data are not publicly available due to restrictions concerning privacy.

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
