# Peer review of "Riociguat in Patients with CTEPH and Advanced Age and/or Comorbidities"

_jcm, 2022, doi:10.3390/jcm11041084_

Round 1

Reviewer 1 Report

Barnikel et al. have investigated the influence of age and comorbidities to the efficacy, tolerability, and side-effects of riociguat. Since previous large studies have excluded older patients and patients with much comorbidities, present study reported the real-world treatment data in patients with CTEPH.

However, there is some limitations, such as the small study group.

Major concerns

  1. Please clarify the definition of side-effects, such as hypotension, and the criteria of dosage reduction of riociguat. Figure 2 showed the no >80 year-old patients with reduced dosage, whereas figure 3 indicated there were hypotension in >80 year-old patients.
  2. The authors showed no statistical difference in the prevalence of side-effects and detailed side-effects between <2 and >2 risk factors. Please discuss the association of the prevalence of side-effects and every comorbidity more detailly.

Minor concerns

  1. Page 5, line 155-158. Please add the data (%) and statistical result of figure 2.
  2. Page 9, line 250-252. Please add the reference.

Reviewer 2 Report

C0ngratulation for your work you were able to show significant improvement in RHC in elderly patients treated with riociguat which was well tolerated. Decrease of mPAP is significant, but numerically it is rather small - my question is how  was systemic pressure  those patients which correlates with mPAP(lower sysytemc pressure - lower mPAP0 - considering hypotensive effect riociguat.

There are some minor correction in table I to be done, why in baseline you included partints wit pulmonary endartectomy and baloon angioplasty, and please look at % in WHO-FC they should not suumarize more  then 100.

Reviewer 3 Report

Barnikel et al. expressed the real-world data of CTEPH treated by riociguat, in a single center.

And the aged group and patients with more HFpEF risk factors showed both tolerability and significant improvement.

【Minor comment】

  1. Patients’ maintenance dose is one of the most critical information for the readers. In addition, the fact that 94% in total and 100% of ≧80years old patients reached the maximum dose(7.5mg/day) was quite important because it can be a selection bias and must have contributed to the improved data after the treatment, as you mentioned. A real-world data of PAH also showed patients reached 6-7.5mg of rociguat was 65.3-74.7% (Hoeper et al. Respiratory medicine 2021).

So, giving the description in the abstract and discussion part and showing the data in the result section makes this article clearer. For instance, can you add the maintenance dose information in the “Safety” section or Table 3?

  1. Table 1:

・It was difficult to understand the meaning of operability. Please explain the definition of each meaning (For example, what is the difference between Pulmonary endarterectomy and PH post PEA? Does pulmonary endarterectomy mean the patients waiting for the operation, taking riociguat?). And when I added all the numbers, it didn’t match the total number and even didn’t reach the total number (Ex. ≧80 years old, Total number is 10. And operability is 2+3+2+1+0+0=8.)

・Do you have to show FC score in Table1? It may be meaningful when comparing baseline-follow-up data or two patients’ groups with the same numbers.

Round 2

Reviewer 1 Report

The manuscript has been improved according to the reviewer’s comments. However, the number of patients is too small to support their conclusion.
